# Tryptophan Modulatory Role in European Seabass (*Dicentrarchus labrax*) Immune Response to Acute Inflammation under Stressful Conditions

**DOI:** 10.3390/ijms232012475

**Published:** 2022-10-18

**Authors:** Marina Machado, Diogo Peixoto, Paulo Santos, Ana Ricardo, Inês Duarte, Inês Carvalho, Cláudia Aragão, Rita Azeredo, Benjamín Costas

**Affiliations:** 1Centro Interdisciplinar de Investigação Marinha e Ambiental (CIIMAR), 4450-208 Matosinhos, Portugal; 2Instituto de Ciências Biomédicas Abel Salazar (ICBAS), Universidade do Porto, 4200-135 Porto, Portugal; 3Instituto Universitario de Investigación Marina (INMAR), Universidad de Cádiz, 11003 Cádiz, Spain; 4Centro de Ciências do Mar (CCMAR), 8005-139 Faro, Portugal; 5Campus de Gambelas, Universidade do Algarve, 8005-139 Faro, Portugal

**Keywords:** tryptophan, innate immunity, aquaculture, functional diets, stress

## Abstract

The present work aimed to study the role of dietary tryptophan supplementation in modulating the European seabass (*Dicentrarchus labrax*) immune condition during stressful rearing conditions (i.e., 15 days exposure to high density), as well as the immune response to acute inflammation after intraperitoneal injection of a bacterial pathogen. Stress alone did not compromise seabass health indicators. In contrast, a clear peripheral and local inflammatory response was observed in response to the inoculated bacteria. Moreover, exposure to a high stocking density seemed to exacerbate the inflammatory response at early sampling points, compared to fish stocked at a lower density. In contrast, stressed fish presented some immune-suppressing effects on the T-cell surface glycoprotein receptor expressions at a late sampling point following inflammation. Regarding the effects of dietary tryptophan, no changes were observed on seabass immune indicators prior to inflammation, while a small number of immunosuppressive effects were observed in response to inflammation, supporting tryptophan’s role in the promotion of immune-tolerance signals during inflammation. Nonetheless, tryptophan dietary supplementation improved the inflammatory response against a bacterial pathogen during stressful conditions, supported by a reduction of plasma cortisol levels, an up-regulation of several immune-related genes at 48 h, and an inversion of the previously observed, stress-induced T-cell suppression. Finally, the involvement of tryptophan catabolism in macrophages was confirmed by the up-regulation of genes involved in the kynurenine pathway. The present study brings new insights regarding the immune modulatory role of tryptophan during stressful conditions in fish, thus allowing for the development of novel prophylactic protocols during vaccination by intraperitoneal injection in the European seabass.

## 1. Introduction

Nowadays, it is widely accepted that incorporation of functional ingredients in aquafeeds represents an efficient farming tool that combines the desired immune modulatory properties, without neglecting the correct nutritional provisions required for fish normal physiological needs. In this context, it is of utmost importance to unveil the nutraceutical potential of tryptophan in the quest for the definition of new strategies to empower fish in the mitigation of the non-ideal production conditions. Recent works have put into evidence the immune tolerance role of tryptophan dietary supplementation in fish, displaying its potential in the self-restrain mechanisms that compose the immune and stress responses. Machado et al. [1] showed that European seabass (*Dicentrarchus labrax*) fed days a tryptophan supplemented diet for 14 and then submitted to an inflammatory insult presented a decreased cellular response and dropped plasma nitric oxide and bactericidal activity, compared to fish fed a control diet. In another study, Machado et al. [2] showed that dietary tryptophan clearly displayed a role in the attenuation of the immune response, once again inhibiting the inflammatory mechanisms and, to some degree, decreasing seabass disease resistance to *Photobacterium damselae* subsp. *piscicida* (*Phdp*). In fact, Machado et al.’s [3] in vitro observations pointed to a more direct role of tryptophan in the limitation of self-damage in inflammatory scenarios by the modulation of pathways associated to the assemblage of anti-inflammatory machinery and induction of macrophages polarization towards an anti-inflammatory phenotype. Moreover, a scenario of dietary tryptophan deficiency may equally represent an handicap for the European seabass, since it may lead to a rise in plasma cortisol levels in response to infection, which may explain the observed compromised immune cell response and disease resistance to *Phdp* [2].

Tryptophan has also recognized roles in the neuroendocrine system and, therefore, works as a player in the neuroendocrine–immune axis. Lepage et al. [4,5] worked on the topic and observed that rainbow trout (*Oncorhynchus mykiss*) fed for 1 week on a diet supplemented with tryptophan (up to 4× above the control diet) undergo different outcomes, according to their stress state, with serotonin (5-HT) acting either as a trigger or inhibitor of the hypothalamic–pituitary–interregnal axis. Taking the fact that fish showed elevated plasma cortisol levels in response to stress into account, tryptophan-supplemented feed led to a significant reduction in the stress-induced elevation of plasma cortisol. In contrast, tryptophan-supplemented feed offered to undisturbed fish resulted in a slight elevation of plasma cortisol levels. In accordance, in an attempt to explore the links between the immune and neuroendocrine responses, Azeredo et al. [6] observed that an increase in dietary tryptophan (i.e., 4× more tryptophan than the control diet) was beneficial in chronically stressed (i.e., high stocking density) Senegalese sole (*Solea senegalensis*) fed for 38 days. However, undisturbed individuals showed a compromised immune status and lower disease resistance in the latter study.

The outcomes of dietary tryptophan trust on its role as a precursor of the compounds responsible for the modulation of stress coping mechanisms and antioxidant and immune mechanisms [7]. In macrophages, tryptophan catabolism is mediated by indoleamine 2,3-dioxygenase (IDO), which catalyzes the first and rate-limiting step of the tryptophan catabolism along the kynurenine pathway. In fact, IDO directly relies on the availability of tryptophan, and its transcription is induced by inflammatory stimuli, such as interferon-γ and cytokines [8,9]. Afterwards, tryptophan catabolic pathway is responsible for exerting anti-microbial effects by tryptophan depletion from the extracellular environment, thus reducing its availability to microorganisms. Moreover, during the kynurenine pathway, the formation of specific metabolites, such as 3-hydroxykynurenine, 3-hydroxyanthranilic acid, and quinolinic acid, regulate T-cell function, and the same metabolites set up a protector system, acting locally to the removal of superoxide radicals modulating oxidative status, thus creating conditions that favor immune suppression and tolerance [8,10]. Similarly, since both immune and stress responses share common pathways, effector organs, and signalling molecules [11], leucocytes are also sensitive to a wide repertoire of other neuroendocrine effectors, such as glucocorticoids. Consequently, important stress signalling molecules, such as cortisol, can have profound effects on the fish immune response by influencing the balanced successive secretion of pro- and anti-inflammatory cytokines [12,13,14]. Actually, tryptophan works as precursor of relevant functional molecules at the neuroendocrine level. The 5-HT is produced from tryptophan by the rate-limiting enzyme tryptophan hydroxylase [15]. This monoamine, which can simultaneously function as a neurotransmitter in the central nervous system and a paracrine or endocrine signal in the gut and blood [16,17], is able to either induce or inhibit the production of adrenocorticotropic hormone (ACTH) in the pituitary. ACTH is then responsible for the increase or decrease of interrenal cortisol synthesis [4]. 

Considering all previous studies, it could be inferred that the consumption needs and physiological fates of tryptophan seems to be modulated according to both the physiological status of the individuals (e.g., stress response) and the progress of inflammation. Therefore, and taking into account the bi-directional and dependent relationship of both neuro-endocrine and immune responses, dietary tryptophan supplementation may represent a tool in the mitigation of stressful conditions during current farming practices, consequently avoiding unfavorable stress-inducing immune suppressor effects upon an infection episode or an inflammatory inducer. Hence, the present work intends to study the ability of dietary tryptophan supplementation to modulate the immune response to an acute inflammatory stimulus in European seabass juveniles reared under stressful conditions (i.e., high stocking density).

## 2. Results

### 2.1. Response to Stress

Exposure to high stocking density-related stress leads to a decrease in the haematocrit levels, while increased haemoglobin and both MCH and MCHC ratios were observed (Table 1). In addition, stressed individuals presented lower mc2r mRNA levels than those left undisturbed, regardless of the feed regiment and sampling time (Appendix A).

### 2.2. Inflammatory Response

The response to inflammation was perceived by looking at the differences found in time, regardless of any other factor, but also in the differences found in time in undisturbed groups (Ø) alone and Ø groups exclusively fed the CTRL diet.

Inflammation alone was responsible for the decrease of peripheral leucocytes numbers, from 48 h to 72 h, but also for the increase of the MCHC from 24 h to 48 h, as well as peripheral monocytes at 4 h and total peritoneal leucocytes from 24 h to 48 h (Table 1 and Table 2). In addition, the number of neutrophils was found to have increased 4 h after infection, while returning to the pre-infection values after 48 h. Plasma cortisol levels reached a peak at 4 and 24 h after infection (Table 3). Gut SOD, CAT, and peroxidase activities were progressively reduced by the inflammation development (Table 4). A general up-regulation of most analyzed genes in the head kidney was found in response to inflammation, reaching a peak at 48 h, with an overall reestablishment of the initial expression values at the sampling point 72 h (Figure 1 and Appendix A). Additionally, the mRNA expression of *cxcr4* (Figure 1C) was found to be higher at 0 and 4 h, compared to the last sampling points, 48 and 72 h, while *il1β* (Figure 1D) showed a higher expression at 4 h, compared to all the sampling points. 

### 2.3. Stress Effect on the Inflammatory Response

The stress outcome on the inflammatory response was assessed taking the response over time, displayed by stressed fish, into account, regardless the feeding regime or whether the fish were fed exclusively the CTRL diet. Additionally, a look was taken at differences displayed between the Ø and stressed groups fed CTRL at a specific sampling point. 

When fish were reared during 15 days in stressful rearing conditions and with the inflammatory mechanisms initiated, a decrease of the haematological indicators, HG, MCH, and MCHC (Table 1), as well as gut SOD, CAT, and tGSH concentration (Table 4), were observed. In fact, in a pre-inflammation status, the abovementioned haematological parameters were found to be increased by stress alone (Section 2.1), which was not observed in undisturbed individuals upon inflammatory induction. Moreover, leucocytes dynamics upon infection were also found to be modulated by the previous stress rearing conditions. Circulating total WBC and neutrophils concentrations were found to be increased at 24 h and 4 h, respectively, in stressed fish, when compared to Ø (Table 2). Curiously, while neutrophils numbers in undisturbed animals rapidly increase in response to inflammation, the number of neutrophils and lymphocytes found in circulation promptly diminished at 4 h in stressed fish, while restoring their initial values in the following sampling times (Table 2). 

Regardless of the diet or whether fish were fed the control feed (CTRL) regime, stress induced a fast up-regulation (mostly found at early sampling points) of several pro-inflammatory genes (Figure 1 and Appendix A), compared the last sampling points, 48 and 72 h. Most importantly, when comparing the response of fish reared under stressful conditions to the undisturbed (Ø) counterparts, when the immune mechanisms were activated, a down-regulation of *tnfα* (Figure 1B), *cd8α* (Figure 1E), *cd3zeta* (Figure 1H), *tcrα* (Figure 1I), *afmid* (Figure 1K), and *ido2* (Figure 1L) at 48 h was observed. In addition, *tgfα* (Figure 1A) was found to be up-regulated by stress at 4 and 24 h. On the other hand, high stocking density led to the up-regulation of *tgfβ* (Figure 1A), *il1β* (Appendix A), *cd8β* (Figure 1F), and *afmid* (Figure 1K) at 4 h, *tgfβ* (Figure 1A) and *tnfα* (Figure 1B) at 24 h, and *tgfβ* (Figure 1A) and *cd8β* (Figure 1F) at 48 h. 

### 2.4. Tryptophan Dietary Supplementation

A look into the tryptophan dietary supplementation effect, in the absence of any other factors, was taken. In addition, tryptophan effect during an inflammatory insult in the absence (Ø) and presence of stress was observed. 

Regardless of the stress condition and inflammatory status, tryptophan dietary supplementation was responsible for a reduction in plasma bactericidal activity (Table 3). Moreover, irrespective to stress, 4 h after infection, the *Phdp* fish fed TRP presented lower MCV (Table 1) and gut SOD concentration (Table 4) than the CTRL-fed fish. Head kidney gene expression showed a down-regulation of *il1β* (Appendix A) at 4 h and *cd8α* (Figure 1E) at 48 h following infection in fish fed TRP, while presenting a higher expression of *tnfα* (Figure 1B) at 24 h than that of the CTRL-fed fish. Moreover, fish fed TRP under stressful rearing conditions reduced cortisol 4 h after inflammation (Table 3), while displaying a general up-regulation of genes such as *tgfβ* (Figure 1A), *cxcr4* (Figure 1C), *cd8α* (Figure 1E), *cd8β* (Figure 1F), *cd3zeta* (Figure 1H), *tcrα* (Figure 1I), *afmid* (Figure 1K), and *ido2* (Figure 1L) at 48 h, compared to CTRL-fed individuals. Finally, the expressions of *tgfβ* (Figure 1A) and *tnfα* (Figure 1B) at 24 h and *cd8β* (Figure 1F) at 4 h were found to be decreased, compared to the CTRL-fed counterparts. 

## 3. Discussion

Aiming to understand the outcome of dietary tryptophan supplementation in an interactive scenario of stressful rearing conditions and an activation of inflammatory mechanisms, European seabass juveniles reared at high stocking density and fed a diet supplemented with tryptophan for 15 days were infected with *Phdp*.

Primarily, stress conditions alone, irrespective of dietary treatment and infection, did not significantly compromise the fish health status, with the exception of a decrease of haematocrit and *mc2r* mRNA expression and an increase of the haemoglobin concentration and corpuscular indexes. The melanocortin 2 receptor (*mc2r*) codes an ACTH receptor that regulates synthesis and release of glucocorticoids in response to ACTH release and, therefore, is essential for the fish stress response. However, the observed down-regulation of the gene, in response to stress, could be the result of a feedback response to the adaptation to prolonged stress [18]. In fact, cortisol levels tended to be higher in stressed European seabass, compared to undisturbed groups at the sampling time 0 h, i.e., after 15 days exposure to high-density conditions. Therefore, as suggested by Agulleiro et al. [18], chronic dietary cortisol may directly result in a decreased MC2R expression as an upstream control mechanism of cortisol release.

Differently, immune stimulation itself induced a clear inflammatory response, as could be expected, that was transversally characterized by a promptly enhanced neutrophil and monocyte response (4 h), followed by an increase of the concentration of total leucocytes found in the peritoneal cavity (from 24 to 48 h). In fact, plasma cortisol levels, as well as the expression of the pro-inflammatory cytokine il1β, were found to be augmented by inflammation right after infection (i.e., 4 and 24 h). Later, the inflammatory process was characterized by the up-regulation of numerous inflammatory genes at 48 h, such as the regulatory cytokine *tgfβ*, the acute-phase cytokine *tnfα*, both T-cell surface glycoproteins *cd8α* and −*β*, and T-cell receptors *cd3zeta* and *tcrα*. The observed modulation of the peripheral and local response to the infection with *Phdp* is in line with several previous reports [2,19,20,21,22] that characterize the acute response to inflammation as a fast neutrophilia and monocytosis within the first 24 h of infection, together with the up-regulation of pro-inflammatory genes at the systemic level. Interestingly, intestinal oxidative stress appeared diminished, as gathered by the decreased antioxidant enzymes superoxide dismutase (SOD), catalase (CAT), and total glutathione (tGSH) activity. This lower activity of antioxidant enzymes at the gut level might be the reflection of a shift in resource allocation to the inflammatory focus [23], the peritoneal cavity, supported by a sturdy response at the peripheral (blood and head kidney genes) and local (peritoneal leucocyte) levels.

Nonetheless, it is worth mentioning that, in the present study, stressful rearing conditions seemed to exacerbate the inflammatory response. Despite the lower haemoglobin, haemoglobin corpuscular indexes, and activity of the antioxidant enzymes SOD, CAT, and tGSH in the gut, compared to non-stressed individuals, stressed fish showed a higher activity of several inflammatory indicators. Contrarily to what would be expected in chronically stressed animals (15 days exposure to high stocking density conditions), no clear innate immune suppression, due to the rise of cortisol [7] levels, seems to have taken place. In fact, stressed seabass clearly displayed a profile resembling the one normally found in acute stressed fish [24]. A significantly higher concentration of peripheral total leucocytes and neutrophils was found in stressed fish at the early stage of inflammation (4 and 24 h). In addition, key inflammatory genes, such as *il1β* and *tgfβ* (at 4 h), *tnfα* (at 24 h) and *cd8α*, and *-β* (at 48 h), were found to be highly expressed in stressed European seabass, compared to non-stressed groups. This points to a probable enhancement of the inflammatory response by stress, both at the leucocyte and transcriptional levels. Nonetheless, stressed fish presented a significantly lower expression of *tnfα*, *cd8α*, *cd3zeta*, and *tcrα* than the undisturbed fish at a late sampling point (48 h). This may, in contrast, suggest a certain degree of immune-suppressing effect of stress, mostly on T-cell surface glycoproteins receptors. In fact, some authors have observed not only a down-regulation of the lymphocyte surface molecules essential for their activation, but also a restrain in their tissue distribution, while the same was not observed in phagocytic cells (neutrophils and monocytes/macrophages) [25,26]. The disparity of the observed results could be the outcome of several factors, such as the type and severity of the stressor used [24,27], despite the prolonged time of exposure applied in the present study (15 days). 

The main goal of the present work was to assess whether dietary tryptophan supplementation during stressful conditions could be of advantage in the event of an infection. 

Exclusively from tryptophan dietary supplementation, apart from a decreased plasma bactericidal activity, no further changes were observed. Such an inability of dietary tryptophan, per se, to induce significant changes in fish immune mechanisms was previously demonstrated in Persian sturgeon (*Acipenser persicus*) by Hoseini et al. [28] and in seabass [1,2] after a 15 days feeding period. Indeed, most of the attributed immune-related effects of dietary tryptophan were described as dependent on immune stimulation and associated with the induction of the IDO-related kynurenine pathway [1,2,8,9,28]. In agreement, a small number of immunosuppressive effects were observed in fish fed a higher tryptophan dose in response to inflammation. Decreases of MCV, gut SOD activity, and down-regulation of the pro-inflammatory cytokine *il1β* and T-cell glycoprotein *cd8α* in the head kidney were observed. Nonetheless, *tnfα* expression was found to be up-regulated at 24 h in response to infection. In a previous study, dietary tryptophan offered 26% above the requirement level led to a weaker peripheral and local leucocyte response, to a down-regulation of key immune-related genes, and to a concomitant decrease in disease resistance to bacterial infection (*Phdp*) [2]. Engelsma et al. [29] also described a similar effect in common carp (*Cyprinus carpio*). Despite being shy, the outcome of the inflammatory response observed in the present study is in agreement with the premise that tryptophan may prime immune suppression by promoting immune-tolerance signals during inflammation [2,3,29,30]. Upon inflammatory induction, tryptophan catabolism in macrophages through the kynurenine pathway contributes to a reduction of its availability for microorganisms consumption, deviates superoxide radicals for the modulation of cell oxidative status, and regulates T-cell function, thus creating conditions that favor immune suppression and tolerance [8,10]. In addition, tryptophan supplementation may contribute to immune tolerance by promoting macrophage differentiation towards the M2/healing phenotype in an inflammatory scenario [3]. Nonetheless, no modulation of either kynurenine-related genes, such as the *afmid* and *ido2*, was observed in response to tryptophan supplementation in an inflammatory scenario. 

In contrast, considering dietary tryptophan supplementation during stressful rearing conditions and its outcome in the inflammatory mechanisms, the present study showed an important reduction of plasma cortisol levels at 4 h of infection, compared to CTRL-fed fish. In fact, despite non-significance, tryptophan seemed to induce a reduction in basal cortisol levels in response to stress, even prior to the inflammatory stimulus (0 h). This is in agreement with the previous reports that point to the ability of tryptophan to counteract stress-induced cortisol increase [4,31,32] and concomitantly avoid the cortisol immune suppressor effect [2,7]. This hypothesis is further supported by the up-regulation of several immune-related genes at 48 h following infection. At that sampling point, increased mRNA expressions of *tgfβ*¸ *cxcr4*, *cd8α*, and *–β*, *cd3zeta* and *tcrα* were observed, highlighting a less restrained response. TGFβ is a multifunctional cytokine produced by leucocytes, with key roles in several cellular process, while CXCR4 is a membrane protein present in leucocytes that binds to the CXC chemokines family. Moreover, contrarily with the previously observed immune-suppressing effect at 48 h of stress, the tryptophan spare seems to revert stress-induced T-cell suppression by increasing the gene expression of the T-cell surface glycoproteins and receptors (*cd8α* and *–β*, *cd3zeta*, *tcrα*). However, it is described that only about 10% of total L-tryptophan is consumed through the neuro-endocrine routes [33]. Indeed, in the present experimental conditions, an up-regulation of *ido2* and *afmid* was visible at 48 h, pointing to the involvement of the tryptophan catabolic pathway in macrophages mediated by IDO. IDO2 is the first and a rate-limiting enzyme of tryptophan catabolism in macrophages, followed by the enzyme arylformamidase (AFMID), which is responsible for kynurenine formation. Through this route, tryptophan consumption sets up a protection system, mostly characterized by its immune suppressive effect and role during the immune response [3,8,10]. Nonetheless, the down-regulation of *tnfα*, *cd8β*, and *tgfβ* at earlier sampling points was perceived. 

To conclude, and despite of the main goal of the present study, the multiplicity of factors such as stress, inflammation, and tryptophan, per se, as well as their interactions, cannot be ignored. To begin with, stressful rearing conditions alone did not significantly compromise seabass health indicators, apart from the expectation of a decreased MC2R expression, possibly as a control mechanism of the prolonged cortisol release. In contrast, immune stimulation itself induced, as expected, a clear inflammatory response at the peripheral (blood and head kidney genes) and local (peritoneal leucocyte) levels. However, no clear immune suppression was perceived when chronically stressed fish (15 days under high stocking density conditions) were submitted to an acute inflammation. Contrarily, stress seemed to exacerbate the inflammatory response at early sampling points (up to 24 h). Nonetheless, stressed fish presented some immune-supressing effects, mostly on T-cell surface glycoproteins receptors at a late sampling point (48 h). 

Finally, while tryptophan alone failed to modulate seabass immune indicators, a small number of immunosuppressive effects were observed in response to inflammation. Such results are in agreement with the premise that tryptophan may prime immune suppression by promoting immune-tolerance signals during inflammation. Nonetheless, when stress is a present factor, dietary tryptophan supplementation seems to improve the imposed inflammatory response by mitigating stress effects, since a clear reduction of plasma cortisol levels was promptly observed in TRP-fed fish. This ability of tryptophan to counteract stress-induced cortisol increase is further supported by the up-regulation of several immune-related genes at 48 h of infection and by a reversal of the previously observed stress-induced T-cell suppression. Finally, the involvement of the tryptophan catabolic pathway in macrophages mediated by IDO was confirmed by the up-regulation of genes involved in the kynurenine pathway. Data from the present study could be useful for the development of novel prophylactic protocols during vaccination by intraperitoneal injection in the European seabass, since fish are crowded prior to i.p. injection, and an improved immune response during inflammation could enhance vaccination efficiency.

## 4. Materials and Methods

### 4.1. Experimental Diets

Two plant protein-based diets with the inclusion of fish soluble protein concentrate (5%) for better palatability were formulated and manufactured by Sparos Lda. (Olhão, Portugal). The CTRL diet was formulated to include an indispensable amino acid (AA) profile meeting the ideal pattern stablished for European seabass [18]. According to previous works [1,2,19], an identical diet was formulated to contain L-tryptophan at 0.3% of feed, at the expense of wheat meal (TRP). All diets were manufactured as described by Machado et al. [34]. Dietary AA content was analyzed by ultra-high performance liquid chromatography (UPLC) after acid hydrolysis, as described in Aragão et al. [35]. Tryptophan was determined by UPLC, after alkaline hydrolysis (6 M NaOH and methanesulfonic acid at 110 °C over 22 h in nitrogen-flushed glass vials). Formulation and proximate analysis in presented in Table 5, whereas the total AA profile of the experimental diets is presented in Table 6. 

### 4.2. Phdp Inoculum Preparation

For the inflammatory bacterial challenge, *Phdp* strain PP3, isolated from yellowtail (*Seriola quinqueradiata*; Japan) by Dr Andrew C. Barnes (Marine Laboratory, Aberdeen, UK), was used. Bacteria were routinely cultured at 22 °C in tryptic soy broth (TSB) or tryptic soy agar (TSA) (both from Difco Laboratories) supplemented with NaCl to a final concentration of 2% (*w*/*v*) (TSB-2 and TSA-2, respectively) and stored at −80 °C in TSB-2 supplemented with 15% (*v*/*v*) glycerol. To prepare the inoculum for injection into the fish peritoneal cavities, 100 µL of stocked bacteria were cultured overnight at 22 °C on TSA-2. Exponentially growing bacteria were collected and re-suspended in sterile TSB-2 and adjusted to a final concentration of 5 × 10^7^ colony forming units (cfu) mL^−1^, as confirmed by plating the resulting cultures onto TSA-2 plates and counting cfu and each fish inoculated intraperitoneally (i.p.) with 100 µL (5 × 10^6^ cfu per fish) of the bacterial suspension.

### 4.3. Experimental Design

European seabass juveniles (12.02 ± 2.77 g) were randomly distributed (n = 21–22 per tank) in two independent recirculating seawater systems, composed of a total of 16 tanks (Figure 2), at a density of 5 kg m^−3^ and maintained for a 2-week acclimatization period (temperature 20 ± 0.5 °C; salinity 32 ppt; photoperiod 10:14 h dark: light). Afterwards, by lowering the water level in one of the systems, fish were submitted to a higher density (10 kg m^−3^; stress group) to simulate a stressful rearing condition, while the other system served as control for the stocking density factor (5 kg m^−3^; Ø group). In a completely randomized design (Figure 2 serves only as example of diet distribution), the two dietary treatments (CTRL and TRP) were randomly assigned in quadruplicate tanks in each recirculating seawater system, and fish were fed these diets for 15 days, twice a day, with a daily average ration of 2% of body weight. Samplings were then carried out (n = 2–3 fish per tank, n = 10 per treatment) for the collection of data, regarding the haematological profile, peritoneal cells collection, plasma cortisol, and immune parameters. In addition, gut samples were collected for innate immune and oxidative stress analysis. Finally, head kidney was sampled for gene expression analysis. Afterwards, the remaining fish were intraperitoneally (i.p.) injected with 100 μL of Phdp (5 × 10^7^ cfu mL^−1^). At that point, temperature was increased up to 24 ± 0.5 °C to mimic the natural infection conditions and, similarly to the previous point, fish were sampled (n = 2–3 fish per tank, n = 10 per treatment) after 4, 24, 48, and 72 h. For all the sampling points, fish were euthanized by anaesthetic overdose with 2-phenoxyethanol (1 mL L^−1^, Merck KGaA).

### 4.4. Haematological Parameters

Blood was collected from the caudal vein using heparinized syringes, one part being used for haematological analysis and the remaining centrifuged at 10,000× *g* 10 min at 4 °C, and the plasma was collected, frozen in dry ice, and stored at −80 °C for evaluating innate humoral immune response parameters. 

The haematological profile was conducted according to Machado et al. [1] and comprised total peripheral leucocytes (WBC) and erythrocytes (RBC) counts, as well as haematocrit (HT) and haemoglobin (HG; SPINREACT kit, ref. 1001230, Spain) assessments. Afterwards, the mean corpuscular volume (MCV), mean corpuscular haemoglobin (MCH), and mean corpuscular haemoglobin concentration (MCHC) were also calculated. Immediately after blood collection, blood smears were performed from homogenized blood and air-dried. After fixation with formol-ethanol (10 of 37% formaldehyde in absolute ethanol), detection of peroxidase was carried out as described by Afonso et al. [19], in order to allow identification of neutrophils. Blood smears were then stained with Wright’s stain (haemacolor; Merck). Slides were examined (×1000 magnification), and at least 200 leucocytes were counted and classified as peripheral thrombocytes, lymphocytes, monocytes, and neutrophils. Absolute value (×10^4^ mL^−1^) of each cell type calculated according to the total blood WBC count.

### 4.5. Peritoneal Leucocytes Counting

Peritoneal cells were collected according to the procedure described by Costas et al. [36] in individuals i.p.-injected with the bacteria. Briefly, following fish anaesthesia and blood collection from the caudal vein, 5 mL of cold Hank’s balanced salt solution (HBSS) supplemented with 30 units heparin mL^−1^ was injected into the peritoneal cavity. The peritoneal area was then slightly massaged, in order to disperse peritoneal cells in the injected HBSS. The i.p.-injected HBSS containing suspended cells were collected, and total peritoneal leucocytes counts were performed with a haemocytometer.

### 4.6. Plasma Innate Immune Response Parameters

Total peroxidase activity in plasma was measured following the procedure described by Quade and Roth [37]. The bactericidal activity assay was performed using *Phdp* strain PP3. Bacteria were cultured in tryptic soy broth (TSB) (Difco Laboratories) supplemented with NaCl to a final concentration of 2% (*w*/*v*) (TSB-2), and exponentially growing bacteria were resuspended in sterile HBSS and adjusted to 1 × 10^6^ cfu mL^−1^. Plating serial dilutions of the suspensions onto TSA-2 plates and counting the number of cfu, following incubation at 22 °C, confirmed bacterial concentration of the inoculum. Plasma bactericidal activity was then determined following the method described by Graham et al. [38] with modifications [1]. 

Cortisol was assessed by an ELISA kit (IBL International Gmbh, Hamburg, Germany) already validated for European seabass [39] and following manufacturer’s instructions. All analyses were conducted in triplicates, and absorbance was read in a Synergy HT (Biotek) microplate reader.

### 4.7. Gut Innate Immune and Oxidative Stress Parameters

Seabass anterior gut samples were weighed and homogenized individually, as described Peixoto et al. [40]. 

Total protein concentration in gut was measured using a Pierce BCA protein assay kit, as described by Costas et al. [41]. Reduced/oxidized glutathione ratio was measured using a microplate assay for GSH/GSSG Kit, as described by Hamre et al. [42]. Catalase (CAT) activity was determined by measuring the decrease of hydrogen peroxide (H_2_O_2_) concentration, as described by Rodrigues et al. [43]. Superoxide dismutase (SOD) activity was determined following Lima et al. [44]. Gut bactericidal activity were determined following the method described by Graham et al. [38] with modifications [40] using *Tenacibaculum maritimum* strain ACC13.1. Peroxidase activity was analyzed according to Quade and Roth [37]. 

All analyses were conducted in triplicates, and absorbance was read in a Synergy HT (Biotek) microplate reader.

### 4.8. Head Kidney Gene Expression

Total RNA isolation, DNase treatment (NZY Total RNA isolation kit, MB13402, NZYTech, Portugal), and first-strand cDNA synthesis (NZY First-strand cDNA synthesis kit, MB125, NZYTech, Portugal) were performed according to manufacture guidelines. Primer design and efficiency values and quantitative PCR assays were performed as described by Machado et al. [20]. Efficiency of each primer was calculated according to a series dilution of a cDNA pool. DNA amplification was carried out using in a CFX384 Touch Real-Time PCR Detection System (Biorad), with specific primers for genes that have been selected for their involvement in immune responses (Table 3). Accession number, efficiency values, annealing temperature, product length, and primers sequences are presented in Table 3. Melting curve analysis was also performed to verify that no primer dimers were amplified. The standard cycling conditions were 95 °C initial denaturation for 10 min, followed by 40 cycles of 94 °C denaturation for 30 s, primer annealing temperature (Table 7) for 30 s, and 72 °C extension for 30 s. All reactions were carried out as technical duplicates. The expression of the target genes was normalized using the expression of European seabass 40s ribosomal protein (40s) and elongation factor 1 alpha (ef1α).

### 4.9. Statistical Analysis

All results are expressed as mean ± standard deviation (mean ± SD). Data were analyzed for normality (Shapiro–Wilk’s W test) and homogeneity of variance (Levene’s test) and, when necessary, transformed before being treated statistically. Data were analyzed by multifactorial ANOVA, with diet, stress, and time/inflammation as factors, and followed by Tukey post-hoc test to identify differences between the experimental treatments. All statistical analyses were performed using the computer package STATISTICA 12 for WINDOWS. The level of significance used was *p* ≤ 0.05 for all statistical tests.

## Figures and Tables

**Figure 1 ijms-23-12475-f001:**
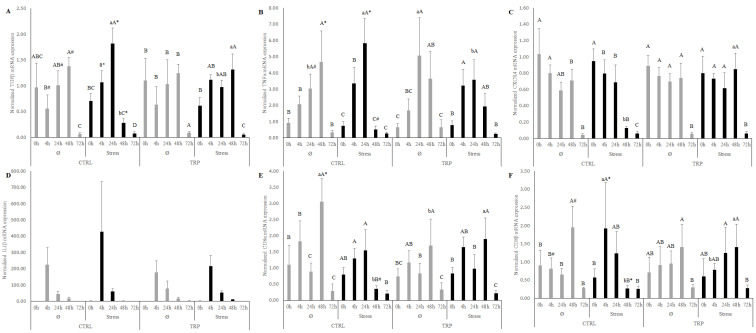
Quantitative expression of (**A**) transforming growth factor-beta, (**B**) tumor necrosis factor alpha, (**C**) chemokine CXC receptor 4, (**D**) interleukin 1 beta, (**E**) cluster of differentiation 8 alpha chain, (**F**) cluster of differentiation 8 beta chain in head kidney of European seabass reared at high density, fed dietary treatments for 15 days (0 h), and sampled 4, 24, 48, and 72 h post-bacterial challenge. Quantitative expression of (**G**) T-cell surface glycoprotein cd4, (**H**) T-cell surface glycoprotein cd3 zeta chain, (**I**) T-cell receptor alpha chain, (**J**) melanocortin 2 receptor, (**K**) arylformamidase-like and (**L**) indoleamine dioxygenase 2 in head kidney of European seabass reared at high density, fed dietary treatments for 15 days (0 h), and sampled 4, 24, 48, and 72 h post-bacterial challenge. Values represent means ± SD (n = 10). Different symbols stand for statistically significant differences attributed to stress. Capital letters stand for statistically significant differences attributed to sampling time. Lower case letters stand for significant differences attributed to dietary treatment (multifactorial ANOVA; Tukey post-hoc test; ns: non-significant; *p* ≤ 0.05).

**Figure 2 ijms-23-12475-f002:**
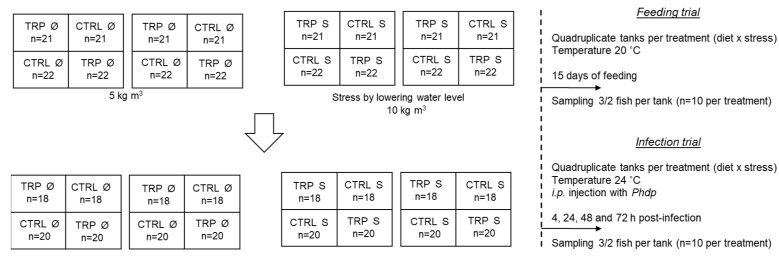
Trial design.

**Table 1 ijms-23-12475-t001:** Haematological parameters of European seabass reared at high density, fed dietary treatments for 15 days (0 h), and sampled 4, 24, 48, and 72 h post-bacterial challenge.

**Parameters**		**Dietary Treatments**																				
		**CTRL**																				
		**Ø**										**Stress**										
		**0 h**	**4 h**			**24 h**		**48 h**	**72 h**			**0 h**	**4 h**				**24 h**		**48 h**		**72 h**	
RBC	(×10^6^ µL^−1^)	1.66 ± 0.20	1.35 ± 0.35			1.45 ± 0.29		1.73 ± 0.12	1.74 ± 0.28			1.33 ± 0.19	1.64 ± 0.50				1.72 ± 0.32		1.83 ± 0.25		1.74 ± 0.13	
HT	(%)	18.17 ± 2.73	21.50 ± 2.36			23.50 ± 3.95		21.17 ± 1.95	19.33 ± 4.23			12.80 ± 1.17	21.50 ± 3.59				21.33 ± 5.28		20.17 ± 3.89		16.83 ± 1.21	
HG	(g dl^−1^)	0.20 ± 0.10	0.09 ± 0.02			0.24 ± 0.09		0.30 ± 0.09	0.11 ± 0.04			0.69 ± 0.41	0.09 ± 0.02				0.08 ± 0.02		0.34 ± 0.11		0.16 ± 0.08	
MCH	(pg cell^−1^)	1.20 ± 0.64	0.79 ± 0.37			1.66 ± 0.57		1.73 ± 0.57	0.67 ± 0.25			5.35 ± 3.38	0.62 ± 0.18				0.46 ± 0.13		1.97 ± 0.80		0.90 ± 0.41	
MCV	(µ m^3^)	110.01 ± 12.03	169.65 ± 42.08			164.65 ± 25.36		102.34 ± 45.65	110.77 ± 13.03			100.93 ± 7.72	140.30 ± 41.05				123.60 ± 14.99		110.83 ± 15.77		97.43 ± 11.64	
MCHC	(g 100 mL^−1^)	1.11 ± 0.60	0.45 ± 0.12			1.05 ± 0.40		4.04 ± 5.60	0.64 ± 0.35			5.22 ± 3.54	0.44 ± 0.09				0.39 ± 0.12		1.76 ± 0.62		1.00 ± 0.51	
**Parameters**		**Dietary treatments**																				
		**TRP**																				
		**Ø**										**Stress**										
		**0 h**	**4 h**			**24 h**		**48 h**	**72 h**			**0 h**	**4 h**				**24 h**		**48 h**		**72 h**	
RBC	(×10^6^ µL^−1^)	1.57 ± 0.26	1.77 ± 0.39			2.06 ± 0.68		1.71 ± 0.31	1.84 ± 0.33			1.50 ± 0.25	1.55 ± 0.40				1.72 ± 0.42		1.86 ± 0.26		1.66 ± 0.40	
HT	(%)	18.83 ± 5.01	19.00 ± 1.91			21.33 ± 4.19		20.83 ± 2.81	17.40 ± 2.58			18.33 ± 1.37	18.83 ± 5.18				18.67 ± 3.50		19.40 ± 1.62		14.67 ± 1.11	
HG	(g dl^−1^)	0.31 ± 0.06	0.11 ± 0.03			0.23 ± 0.04		0.29 ± 0.08	0.21 ± 0.23			0.66 ± 0.46	0.10 ± 0.02				0.12 ± 0.07		0.29 ± 0.06		0.24 ± 0.11	
MCH	(pg cell^−1^)	1.98 ± 0.33	0.67 ± 0.22			1.26 ± 0.45		1.59 ± 0.78	1.16 ± 1.17			4.61 ± 2.26	0.65 ± 0.18				0.68 ± 0.36		1.57 ± 0.33		1.53 ± 0.79	
MCV	(µ m^3^)	120.77 ± 28.73	111.42 ± 22.61			109.38 ± 23.21		116.02 ± 6.09	97.10 ± 8.77			124.52 ± 17.00	124.97 ± 38.85				113.97 ± 29.44		107.95 ± 13.53		92.83 ± 21.66	
MCHC	(g 100 mL^−1^)	1.72 ± 0.42	0.61 ± 0.22			1.17 ± 0.45		1.86 ± 0.60	0.67 ± 0.28			3.53 ± 2.40	0.56 ± 0.22				0.65 ± 0.41		1.55 ± 0.34		1.66 ± 0.83	
**Multifactorial ANOVA**																						
**Parameters**	**Diet**	**Stress**	**Time**			**Diet × Stress**			**Diet × Time**			**Stress × Time**	**Diet × Stress × Time**				**Time**					
																	**0 h**		**4 h**	**24 h**	**48 h**	**72 h**
RBC	ns	ns	ns			ns			ns			ns	ns									
HT	ns	0.048	0.006			ns			0.035			ns	ns				B		AB	A	AB	B
HG	ns	ns	<0.001			ns			ns			<0.001	ns				A		C	C	B	BC
MCH	ns	0.045	<0.001			ns			ns			<0.001	ns				A		C	B	B	BC
MCV	0.032	ns	<0.001			ns			0.004			ns	ns				ABC		A	AB	B	C
MCHC	ns	ns	<0.001			ns			ns			0.001	ns				A		C	C	B	BC
**Parameters**			**Diet × Time**										**Stress × Time**									
			**CTRL**					**TRP**					**Ø**				**Stress**					
			**0 h**	**4 h**	**24 h**	**48 h**	**72 h**	**0 h**	**4 h**	**24 h**	**48 h**	**72 h**	**0 h**	**4 h**	**24 h**	**48 h**	**72 h**	**0 h**	**4 h**	**24 h**	**48 h**	**72 h**
HT			B	A	A	AB	AB															
HG													#					A *	B	B	B	B
MCH													#					A *	B	B	B	B
MCV			B	aA	A	B	B		b													
MCHC													AB #	AB	B	A	AB	A *	B	B	B	B

Values represent means ± SD (n = 10). Different symbols stand for statistically significant differences attributed to stress. Capital letters stand for statistically significant differences attributed to sampling time. Lower case letters stand for significant differences attributed to dietary treatment (multifactorial ANOVA; Tukey post-hoc test; ns: non-significant; *p* ≤ 0.05). RBC (red blood cells-erythrocytes); Ht (haematocrit); Hg (haemoblogin); MCH (mean corpuscular haemoglobin); MCV (mean corpuscular volume); MCHC (mean corpuscular haemoglobin concentration).

**Table 2 ijms-23-12475-t002:** Peripheral and peritoneal leucocyte counts of European seabass reared at high density, fed dietary treatments for 15 days (0 h), and sampled 4, 24, 48, and 72 h post-bacterial challenge.

**Parameters**		**Dietary Treatments**																				
		**CTRL**																				
		**Ø**									**Stress**											
		**0 h**		**4 h**	**24 h**		**48 h**	**72 h**			**0 h**		**4 h**			**24 h**		**48 h**		**72 h**		
WBC	(×10^4^ µL^−1^)	5.02 ± 0.88		4.13 ± 1.00	4.53 ± 0.92		5.03 ± 0.96	4.02 ± 0.42			3.80 ± 1.00		4.50 ± 0.93			4.85 ± 0.69		4.73 ± 0.46		4.83 ± 1.32		
Peripheral Neutrophils		0.03 ± 0.03		0.34 ± 0.17	0.40 ± 0.05		0.09 ± 0.05	0.05 ± 0.04			0.04 ± 0.05		0.50 ± 0.20			0.30 ± 0.10		0.05 ± 0.02		0.06 ± 0.05		
Peripheral Monocytes		0.09 ± 0.02		0.18 ± 0.11	0.08 ± 0.05		0.25 ± 0.13	0.11 ± 0.06			0.05 ± 0.03		0.21 ± 0.11			0.18 ± 0.10		0.14 ± 0.04		0.21 ± 0.19		
Peripheral Lymphocytes		1.46 ± 0.44		0.83 ± 0.24	1.40 ± 0.35		1.31 ± 0.44	1.08 ± 0.26			1.19 ± 0.27		0.53 ± 0.20			1.16 ± 0.47		1.53 ± 0.35		1.54 ± 0.44		
Peripheral Thrombocytes		3.51 ± 0.67		3.08 ± 0.85	2.70 ± 0.60		3.39 ± 0.64	2.79 ± 0.47			2.76 ± 0.78		3.30 ± 0.71			3.36 ± 0.31		2.85 ± 0.35		3.04 ± 0.97		
Peritoneal Leucocytes		-		5.77 ± 4.51	4.40 ± 2.15		7.88 ± 3.97	6.68 ± 2.16			-		6.58 ± 4.22			4.87 ± 1.23		7.00 ± 2.33		6.25 ± 2.11		
**Parameters**		**Dietary treatments**																				
		**TRP**																				
		**Ø**									**Stress**											
		**0 h**		**4 h**	**24 h**		**48 h**	**72 h**			**0 h**		**4 h**			**24 h**		**48 h**		**72 h**		
WBC	(×10^4^ µL^−1^)	4.37 ± 1.31		3.93 ± 0.82	4.02 ± 0.86		4.57 ± 0.63	3.95 ± 0.42			5.08 ± 0.89		4.42 ± 1.46			5.53 ± 1.03		3.33 ± 0.61		4.73 ± 0.83		
Peripheral Neutrophils		0.03 ± 0.03		0.22 ± 0.14	0.40 ± 0.19		0.09 ± 0.03	0.07 ± 0.10			0.12 ± 0.14		0.49 ± 0.30			0.45 ± 0.20		0.05 ± 0.03		0.03 ± 0.02		
Peripheral Monocytes		0.12 ± 0.06		0.21 ± 0.12	0.07 ± 0.03		0.13 ± 0.03	0.14 ± 0.07			0.14 ± 0.14		0.16 ± 0.10			0.18 ± 0.08		0.14 ± 0.05		0.15 ± 0.05		
Peripheral Lymphocytes		1.39 ± 0.49		0.92 ± 0.65	1.34 ± 0.27		1.42 ± 0.36	1.21 ± 0.28			1.35 ± 0.36		0.57 ± 0.28			1.48 ± 0.47		0.99 ± 0.19		1.53 ± 0.44		
Peripheral Thrombocytes		2.85 ± 0.84		3.11 ± 0.66	2.28 ± 0.52		2.31 ± 0.55	2.53 ± 0.37			3.49 ± 0.64		3.18 ± 0.94			3.50 ± 0.88		2.17 ± 0.52		3.08 ± 0.60		
Peritoneal Leucocytes		-		5.60 ± 4.36	5.25 ± 2.69		7.38 ± 6.71	7.53 ± 3.27			-		6.82 ± 3.71			5.20 ± 1.24		7.15 ± 3.16		8.37 ± 3.71		
**Multifactorial ANOVA**																						
**Parameters**	**Diet**	**Stress**	**Time**	**Diet × Stress**	**Diet × Time**	**Stress × Time**	**Diet × Stress × Time**	**Time**					**Stress × Time**									
													**Ø**					**Stress**				
								**0 h**	**4 h**	**24 h**	**48 h**	**72 h**	**0 h**	**4 h**	**24 h**	**48 h**	**72 h**	**0 h**	**4 h**	**24 h**	**48 h**	**72 h**
WBC	ns	ns	ns	ns	ns	0.02	ns						AB	AB	AB #	A	B	AB	AB	A *	B	AB
Peripheral Neutrophils	ns	ns	<0.001	ns	ns	0.011	ns	B	A	A	B	B	B	A #	A	B	B	A	B *	B	A	A
Peripheral Monocytes	ns	ns	0.025	ns	ns	ns	ns	B	A	AB	AB	AB										
Peripheral Lymphocytes	ns	ns	<0.001	ns	ns	0.048	ns	A	B	A	A	A						A	B	A	A	A
Peripheral Thrombocytes	ns	ns	ns	ns	ns	ns	ns															
Peritoneal Leucocytes	ns	ns	0.045	ns	ns	ns	ns	-	AB	B	A	AB										

Values represent means ± SD (n = 10). Different symbols stand for statistically significant differences attributed to stress. Capital letters stand for statistically significant differences attributed to sampling time. (Multifactorial ANOVA; Tukey post-hoc test; ns: non-significant; *p* ≤ 0.05).

**Table 3 ijms-23-12475-t003:** Plasma humoral parameters of European seabass reared at high density, fed dietary treatments for 15 days (0 h), and sampled 4, 24, 48, and 72 h post-bacterial challenge.

**Parameters**		**Dietary Treatments**																				
		**CTRL**																				
		**Ø**							**Stress**													
		**0 h**		**4 h**	**24 h**	**48 h**		**72 h**	**0 h**		**4 h**		**24 h**				**48 h**			**72 h**		
Peroxidase	(unit mL^−1^)	25.70 ± 0.00		39.71 ± 18.83	26.42 ± 23.86	44.30 ± 31.57		107.03 ± 50.21	89.27 ± 65.83		46.51 ± 5.59		70.31 ± 6.54				62.25 ± 31.51			76.80 ± 37.97		
Bactericidal Activity	(%)	20.48 ± 7.49		24.84 ± 8.98	21.20 ± 4.96	24.04 ± 7.42		16.12 ± 5.05	15.71 ± 8.84		27.53 ± 17.83		28.66 ± 9.61				23.48 ± 10.28			26.28 ± 18.85		
Cortisol	(ng mL^−1^)	83.82 ± 12.13 B		157.52 ± 38.29 A	170.42 ± 30.29 A	119.35 ± 39.47 AB		68.36 ± 28.77 B	103.76 ± 15.91 AB		174.00 ± 28.08 aA		116.51 ± 19.67 AB				119.35 ± 36.04 AB			80.93 ± 16.47 B		
**Parameters**		**Dietary treatments**																				
		**TRP**																				
		**Ø**							**Stress**													
		**0 h**		**4 h**	**24 h**	**48 h**		**72 h**	**0 h**		**4 h**		**24 h**				**48 h**			**72 h**		
Peroxidase	(unit mL^−1^)	74.24 ± 46.42		42.68 ± 12.26	62.33 ± 30.93	69.64 ± 77.44		63.90 ± 48.50	28.10 ± 1.61		60.03 ± 29.00		82.57 ± 66.71				115.68 ± 69.28			64.74 ± 37.44		
Bactericidal Activity	(%)	25.26 ± 14.05		21.81 ± 9.09	22.00 ± 11.10	19.24 ± 6.39		13.67 ± 10.28	15.49 ± 7.01		19.46 ± 4.97		11.40 ± 13.65				22.61 ± 11.10			13.59 ± 4.82		
Cortisol	(ng mL^−1^)	127.452 ± 42.81 AB		152.30 ± 26.32 AB	174.00 ± 30.76 A	80.93 ± 18.05 B		62.19 ± 27.96 B	74.87 ± 33.28		87.77 ± 17.47 b		116.51 ± 21.99				98.20 ± 24.74			48.72 ± 20.18		
**Multifactorial ANOVA**																						
		**Diet**	**Stress**	**Time**	**Diet × Stress**	**Diet × Time**	**Stress × Time**	**Diet × Stress × Time**	**Diet × Stress**				**Stress × Time**									
									**CTRL**		**TRP**		**Ø**					**Stress**				
									**Ø**	**Stress**	**Ø**	**Stress**	**0 h**	**4 h**	**24 h**	**48 h**	**72 h**	**0 h**	**4 h**	**24 h**	**48 h**	**72 h**
Peroxidase		ns	ns	ns	ns	ns	ns	ns					B	B	B	AB	A					
Bactericidal Activity		0.033	ns	ns	ns	ns	ns	ns														
Cortisol		ns	<0.001	ns	<0.001	ns	<0.001	0.003	b	a	A *	B #				*	*	A	AB	AB	AB #	B #

Values represent means ± SD (n = 10). Different symbols stand for statistically significant differences attributed to stress. Capital letters stand for statistically significant differences attributed to sampling time. Lower case letters stand for significant differences attributed to dietary treatment (multifactorial ANOVA; Tukey post-hoc test; ns: non-significant; *p* ≤ 0.05).

**Table 4 ijms-23-12475-t004:** Gut immune and oxidative stress parameters of European seabass reared at high density, fed dietary treatments for 15 days (0 h), and sampled 4, 24, 48, and 72 h post-bacterial challenge.

**Parameters**			**Dietary Treatments**																													
			**CTRL**																													
			**Ø**										**Stress**																			
			**0 h**		**4 h**	**24 h**		**48 h**			**72 h**		**0 h**				**4 h**					**24 h**			**48 h**				**72 h**			
SOD	(U mg protein^−1^)		42.50 ± 16.84		55.84 ± 29.61	15.38 ± 4.40		23.51 ± 9.56			17.80 ± 3.54		53.95 ± 35.15				57.60 ± 31.57					58.37 ± 28.88			24.37 ± 8.21				13.03 ± 6.39			
CAT	(U mg protein^−1^)		57.68 ± 32.03		87.07 ± 35.31	29.76 ± 0.00		47.41 ± 16.13			40.62 ± 1.65		83.74 ± 17.14				107.67 ± 23.65					59.68 ± 6.75			37.31 ± 7.74				45.53 ± 10.19			
Total GSH	(μM)		2321.68 ± 749.79 A		3497.65 ± 1490.63 A *	845.78 ± 296.25 B		226.61 ± 77.26 B			283.05 ± 81.95 B		2834.11 ± 790.40 A				1290.81 ± 648.55 B #					895.02 ± 347.87 B			357.92 ± 49.20 B				318.32 ± 86.37 B			
Bactericidal Activity	(%)		58.76 ± 6.42		64.83 ± 7.06	66.50 ± 2.90		67.64 ± 1.70			66.59 ± 1.79		59.55 ± 5.18				62.37 ± 2.70					60.01 ± 6.70			68.35 ± 4.65				70.07 ± 3.13			
Peroxidase	(unit mL^−1^)		28.13 ± 8.60		77.31 ± 78.10	65.66 ± 55.01		89.85 ± 47.17			23.83 ± 4.05		44.34 ± 15.57				56.56 ± 26.38					58.36 ± 34.05			66.72 ± 47.22				21.49 ± 7.53			
**Parameters**			**Dietary treatments**																													
			**TRP**																													
			**Ø**										**Stress**																			
			**0 h**		**4 h**	**24 h**		**48 h**			**72 h**		**0 h**				**4 h**					**24 h**			**48 h**				**72 h**			
SOD	(U mg protein^−1^)		59.19 ± 23.38		21.52 ± 15.67	17.24 ± 9.07		36.20 ± 15.68			11.82 ± 8.48		44.55 ± 31.92				29.15 ± 10.78					30.65 ± 25.24			14.67 ± 7.66				11.85 ± 5.40			
CAT	(U mg protein^−1^)		73.92 ± 18.86		68.06 ± 11.05	44.33 ± 8.15		60.96 ± 12.10			41.25 ± 10.45		66.95 ± 10.48				90.37 ± 32.22					63.27 ± 2.30			38.06 ± 4.43				31.68 ± 1.47			
Total GSH	(μM)		2778.96 ± 305.04 A		2839.77 ± 846.36 A	753.84 ± 723.88 B		440.67 ± 108.38 B			354.49 ± 154.28 B		2013.46 ± 460.27 AB				2608.92 ± 1299.99 A					681.46 ± 221.56 BC			546.07 ± 46.62 BC				312.23 ± 170.54 C			
Bactericidal Activity	(%)		60.43 ± 5.61 B		65.01 ± 2.65 AB	60.43 ± 3.52 AB		67.85 ± 1.72 AB			72.50 ± 2.45 A		66.91 ± 4.18 AB				58.59 ± 4.33 B					63.92 ± 1.39 AB			69.37 ± 3.09 A				66.91 ± 4.83 AB			
Peroxidase	(unit mL^−1^)		54.70 ± 14.38		50.71 ± 31.48	44.33 ± 27.21		52.12 ± 27.78			31.24 ± 12.96		77.31 ± 45.37				69.84 ± 30.96					33.71 ± 16.23			64.88 ± 39.43				26.55 ± 7.58			
**Multifactorial ANOVA**																																
	**Diet**	**Stress**	**Time**	**Diet × Stress**	**Diet × Time**	**Stress × Time**	**Diet × Stress × Time**	**Time**					**Diet × Time**										**Stress × Time**									
													**CTRL**					**TRP**					**Ø**					**Stress**				
								**0 h**	**4 h**	**24 h**	**48 h**	**72 h**	**0 h**	**4 h**	**24 h**	**48 h**	**72 h**	**0 h**	**4 h**	**24 h**	**48 h**	**72 h**	**0 h**	**4 h**	**24 h**	**48 h**	**72 h**	**0 h**	**4 h**	**24 h**	**48 h**	**72 h**
SOD	0.037	ns	<0.001	ns	0.044	0.032	ns	A	B	BC	BC	C	A	AB*	ABC	B	C		#				A	AB	B	AB	B	A	B	B	BC	B
CAT	ns	ns	<0.011	ns	ns	0.018	ns	AB	A	BC	C	C											AB	A	B	AB	B	A	A	BC	C	C
Total GSH	ns	ns	<0.001	ns	ns	0.009	0.005	A	A	B	B	B											A	A *	B	B	B	A	A #	B	B	B
Bactericidal Activity	ns	ns	<0.001	ns	ns	ns	0.021	B	B	B	A	A																				
Peroxidase	ns	ns	0.03	ns	ns	ns	ns	AB	A	AB	A	B																				

Values represent means ± SD (n = 10). Different symbols stand for statistically significant differences attributed to stress. Capital letters stand for statistically significant differences attributed to sampling time (multifactorial ANOVA; Tukey post-hoc test; ns: non-significant; *p* ≤ 0.05).

**Table 5 ijms-23-12475-t005:** Ingredients and chemical composition of the experimental diets.

Ingredients (% Feed)	CTRL	TRP
Fish soluble protein concentrate ^a^	5.00	5.00
Fish gelatin ^b^	2.00	2.00
Soy protein concentrate ^c^	25.00	25.00
Pea protein concentrate ^d^	6.00	6.00
Wheat gluten ^e^	10.00	10.00
Corn gluten meal ^f^	15.00	15.00
Wheat meal ^g^	15.80	15.50
Fish oil ^h^	15.10	15.10
Vit and min premix ^i^	1.00	1.00
Soy lecithin ^j^	1.00	1.00
Antioxidant ^k^	0.20	0.20
Sodium propionate ^l^	0.10	0.10
Monocalcium phosphate ^m^	3.00	3.00
L-lysine ^n^	0.60	0.60
L-tryptophan ^o^	0.00	0.30
DL-methionine ^p^	0.20	0.20
Total	100.00	100.00
Chemical composition (as fed basis)		
Crude protein (%)	45.70	46.00
Crude fat (%)	18.00	18.00
Fiber (%)	1.70	1.70
Starch (%)	13.40	13.20
Ash (%)	6.80	6.80
Energy (MJ/kg)	21.90	21.90

^a^ CPSP 90: 82.6% crude protein (CP), 9.6% crude fat (CF), Sopropêche, France. ^b^ Fish gelatin: 88% CP, 0.1% CF, LAPI Gelatine SPA, Italy. ^c^ Soycomil P: 63% CP, 0.8% CF, ADM, The Netherlands. ^d^ NUTRALYS F85F: 78% CP, 1% CF, ROQUETTE, Frères, France. ^e^ VITAL: 83.7% CP, 1.6% CF, ROQUETTE, Frères, France. ^f^ Corn gluten meal: 61% CP, 6% CF, COPAM, Portugal. ^g^ Wheat meal: 10.2% CP; 1.2% CF, Casa Lanchinha, Portugal. ^h^ SAVINOR UTS, Portugal. ^I^ PREMIX Lda, Portugal. Vitamins (IU or mg/kg diet): DL-alpha tocopherol acetate, 100  mg; sodium menadione bisulphate, 25  mg; retinyl acetate, 20,000 IU; DL-cholecalciferol, 2000 IU; thiamin, 30  mg; riboflavin, 30  mg; pyridoxine, 20  mg; cyanocobalamin, 0.1  mg; nicotinic acid, 200  mg; folic acid, 15  mg; ascorbic acid, 500  mg; inositol, 500  mg; biotin, 3  mg; calcium panthotenate, 100  mg; choline chloride, 1000  mg, betaine, 500  mg. Minerals (g or mg/kg diet): copper sulphate, 9  mg; ferric sulphate, 6  mg; potassium iodide, 0.5  mg; manganese oxide, 9.6  mg; sodium selenite, 0.01  mg; zinc sulphate, 7.5  mg; sodium chloride, 400  mg; excipient wheat middlings. ^j^ Lecico P700IPM, LECICO GmbH, Germany. ^k^ Paramega PX, Kemin Europe NV, Belgium. ^l^ PREMIX Lda., Portugal. ^m^ MCP: 22% phosphorus, 16% calcium, Fosfitalia, Italy. ^n^ Lysine HCl 99%, Ajinomoto Eurolysine SAS, France. ^o^ L-Tryptophan 98%, Ajinomoto Eurolysine SAS, France. ^p^ DL-methionine for aquaculture: 99% methionine, Evonik Nutrition and Care GmbH, Germany.

**Table 6 ijms-23-12475-t006:** Amino acid composition of experimental diets.

Amino Acids (% DW)	CTRL	TRP
Arginine	4.0	4.1
Histidine	1.2	1.2
Lysine	3.0	3.0
Threonine	1.8	1.9
Isoleucine	2.1	2.1
Leucine	4.2	4.3
Valine	2.3	2.3
Methionine	1.1	1.2
Phenylalanine	3.0	3.2
Cysteine	0.4	0.4
Tyrosine	2.6	2.8
Aspartic acid	3.4	3.2
Glutamic acid	9.7	9.1
Alanine	2.4	2.5
Glycine	2.7	2.7
Proline	3.7	3.6
Serine	2.6	2.6
Tryptophan	**0.2**	**0.4**

**Table 7 ijms-23-12475-t007:** Forward and reverse primers for real-time PCR.

Acronym	Gene	Gene Bank ID	Eff ^1^	AT ^2^	Product Length ^3^	Forward Primer Sequence (5′-3′)	Reverse Primer Sequence (5′-3′)
*40s*	40s ribosomal protein	HE978789.1	104.44	60	79	TGATTGTGACAGACCCTCGTG	CACAGAGCAATGGTGGGGAT
*ef1α*	Elongation factor 1 β	AJ866727.1	96.45	57	144	AACTTCAACGCCCAGGTCAT	CTTCTTGCCAGAACGACGGT
*tgfβ*	Transforming growth factor β	AM421619.1	109.67	55	143	ACCTACATCTGGAACGCTGA	TGTTGCCTGCCCACATAGTAG
*tnfα*	Tumor necrosis factor α	DQ070246.1	108.81	55	112	AGCCACAGGATCTGGAGCTA	GTCCGCTTCTGTAGCTGTCC
*cxcr4*	Chemokine CXC receptor 4	FN687464.1	90.94	57	171	ACCAGACCTTGTGTTTGCCA	ATGAAGCCCACCAGGATGTG
*il1β*	Interleukin 1 β	AJ269472.1	87.60	57	105	AGCGACATGGTGCGATTTCT	CTCCTCTGCTGTGCTGATGT
*cd8α*	Cluster of differentiation 8 α chain	DQ090838.1	117.69	60	292	GCGATTCTGGTGTTTTGTCTAAAG	GAGACGCACAGCTGTAAATGC
*cd8β*	Cluster of differentiation 8 β chain	DLAgn_00090370	104.44	55	223	CGGAACCCAAAAGGCCAAAG	TAGGCTGTAGATGCAGTGCT
*cd4*	T-cell surface glycoprotein cd4	DLAgn_00058650	98.04	60	131	CTCGGTGGCATCCTTCTGAC	AGCCAGACAACAACCTGTCC
*cd3zeta*	T-cell surface glycoprotein cd3 zeta chain	DLAgn_00052540	93.08	60	99	GGCTCTCCGAACTTCTCCAG	CTGCAACGTCGTTTCCGATG
*tcrα*	T-cell receptor α chain	DLAgn_00260540	116.00	60	72	ACACTGGCTGAGAAACATCCT	GGCTGGGTCACTCTGTCTTC
*mc2r*	Melanocortin 2 receptor	DLAgn_00065140	100.98	60	118	GAGGGCAAGGGGAGCATTTA	GACGGGCAGATGGCAGTTAT
*afmid*	Arylformamidase-like	DLAgn_00177950	128.26	55	112	CGTTTCCACCTGTTTGACCT	CCTAGCCTGCTGAAGGACTG
*ido2*	Indoleamine-dioxygenase 2	DLAgn_00014730	108.20	55	74	TGAAGGTGTGAGCAATGAGC	CAAAGCACTGAATGGCTGAA

^1^ The efficiency of PCR reactions calculated from serial dilutions of tissue RT reactions in the validation procedure. ^2^ Annealing temperature (°C). ^3^ Amplicon (bp).

## Data Availability

All data are provided in the main text or Appendix A.

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
