# Peer review of "Tryptophan Modulatory Role in European Seabass (Dicentrarchus labrax) Immune Response to Acute Inflammation under Stressful Conditions"

_ijms, 2022, doi:10.3390/ijms232012475_

Round 1

Reviewer 1 Report

This a work dealing with the effects of dietary Trp supplements on stress and immune responses in fish. Spite of being a topic which has been studied in other works, the authors provide new insights between effects and interactions on several factors affecting the health status.

The experimental design and cultures development is suitable, taking into account the appropriate application of the different factors influencing the physiological responses.

The use of several factors is a good approach but it always makes difficult the interpretation of results and conclusions extraction. Undoubtedly the presentation of results is complex due to it and the analysis of many variables, and I think authors could improve it to make it more understandable for the reader. I have written some comments on it in the attached file.

Nevertheless I understand that is a very complex work because the results coming from ANOVAs having more than two factors needs to be analysed and revised deeply, to achieve to coherent conclusions. And authors have done it well, though maybe the presentation of data could be improved, as pointed.

Concluding, I think this work could be accepted for publication after some moderate changes (see attached PDF).

Author Response

Dear reviewer,

the authors would like to thank for all corrections to the body of the text. Please find enclosed in the file some comments.

Reviewer 2 Report

Dr. Muhammad Safdar [email protected]

Sat, 2 Oct 2021, 22:29    
to me

Reviewer #1: In this manuscript, the researchers tried to explain about the Tryptophan modulatory role in European seabass (Dicentrarchus labrax) immune response to acute inflammation under stressful conditions. It is interesting work and can be accepted after revision.

-     The grammar errors should be checked in the whole manuscript.

-       In abstract, the first four lines should be summarized.

-       In introduction, the main objective has been repeated so it should be refined.

-       Some recent and relevant articles may be added as thousands of articles have been published on this topic.

-       Conclusion should be refined as it is not properly written as per results.

Author Response

Dear reviewer,
